# Effectiveness of Infection Control Teams in Reducing Healthcare-Associated Infections: A Systematic Review and Meta-Analysis

**DOI:** 10.3390/ijerph192417075

**Published:** 2022-12-19

**Authors:** Moe Moe Thandar, Md. Obaidur Rahman, Rei Haruyama, Sadatoshi Matsuoka, Sumiyo Okawa, Jun Moriyama, Yuta Yokobori, Chieko Matsubara, Mari Nagai, Erika Ota, Toshiaki Baba

**Affiliations:** 1Bureau of International Health Cooperation, National Center for Global Health and Medicine, Tokyo 162-8655, Japan; 2Center for Surveillance, Immunization, and Epidemiologic Research, National Institute of Infectious Diseases, Tokyo 162-8640, Japan; 3Center for Evidence-Based Medicine and Clinical Research, Dhaka 1230, Bangladesh; 4Global Health Nursing, Graduate School of Nursing Sciences, St. Luke’s International University, Tokyo 104-0044, Japan; 5Tokyo Foundation for Policy Research, Minato, Tokyo 106-0032, Japan

**Keywords:** infection control team, infection control link nurse, healthcare-associated infections, systematic review

## Abstract

The infection control team (ICT) ensures the implementation of infection control guidelines in healthcare facilities. This systematic review aims to evaluate the effectiveness of ICT, with or without an infection control link nurse (ICLN) system, in reducing healthcare-associated infections (HCAIs). We searched four databases to identify randomised controlled trials (RCTs) in inpatient, outpatient and long-term care facilities. We judged the quality of the studies, conducted meta-analyses whenever interventions and outcome measures were comparable in at least two studies, and assessed the certainty of evidence. Nine RCTs were included; all were rated as being low quality. Overall, ICT, with or without an ICLN system, did not reduce the incidence rate of HCAIs [risk ratio (RR) = 0.65, 95% confidence interval (CI): 0.45–1.07], death due to HCAIs (RR = 0.32, 95% CI: 0.04–2.69) and length of hospital stay (42 days vs. 45 days, *p* = 0.52). However, ICT with an ICLN system improved nurses’ compliance with infection control practices (RR = 1.17, 95% CI: 1.00–1.38). Due to the high level of bias, inconsistency and imprecision, these findings should be considered with caution. High-quality studies using similar outcome measures are needed to demonstrate the effectiveness and cost-effectiveness of ICT.

## 1. Introduction

Healthcare-associated infections (HCAIs) are infections acquired by patients while receiving care for other illnesses from hospitals, acute care clinics, community health centres or care homes [1]. HCAIs are common in acute care hospitals, affecting 7–10% of patients globally [2]. They also occur in different types of care settings, such as outpatient clinics, ambulatory surgical centres, outpatient dialysis facilities and long-term care facilities (nursing homes and rehabilitation facilities) [3].

Up to 70% of HCAIs are preventable [4,5,6]. In hospitals, infection control departments with dedicated personnel are an important feature of HCAI prevention [7]. To prevent HCAIs, infection control guidelines need to be implemented, which is usually the task of the infection control team (ICT) [8]. An ICT plays important roles in different aspects of infection control, such as (1) developing and disseminating guidelines and policies, (2) coordinating continuous education and training, (3) establishing systems for the surveillance of HCAIs, (4) monitoring and auditing the practices and standards of care and (5) building effective links with other staff and departments [2,9].

The core members of ICT are doctors, epidemiologists, microbiologists and nurses, known as infection control nurses (ICNs). At present, the ICT system has expanded to include new roles, such as infection control link nurses (ICLNs) and infection control champions. ICLNs and infection control champions are ward-based staff working under the supervision of ICNs [10] and acting as a link between their own clinical wards and the ICT [9]. The ICT and ICLN system are also applied in nursing homes [11,12,13,14,15].

ICT was established in the United Kingdom in 1950s and ICLN system was first introduced in 1990s. To date, previous reviews have also reported on ICT work practices [9], concept of the ICLN system [16] and facilitators and barriers in the ICLN system implementation [17]. However, an initial search for literature on infection control revealed the paucity of systematic reviews on the efficacy of ICT, with or without ICLN system, in inpatient hospitals and outpatient and long-term care facilities. In this systematic review, we therefore aimed to evaluate its effectiveness, with or without the ICLN systems, in reducing the rate of HCAIs in inpatient hospitals and outpatient and long-term care facilities.

## 2. Materials and Methods

### 2.1. Protocol and Registration

We registered the review protocol in the International Prospective Register of Systematic Reviews (PROSPERO) (registration number: CRD42020172173) and published it in BMJ Open [18]. This review adheres to the Preferred Reporting Items for Systematic Reviews and Meta-analysis (PRISMA) statement [19] (Appendix A).

### 2.2. Eligibility Criteria

Randomised controlled trials (RCTs) that compared an ICT, with or without the ICLN system, with any other intervention or no intervention were eligible (Appendix A). We included studies on patients of all ages in inpatient hospitals and outpatient facilities, and residents in long-term care facilities to examine the patient-based outcomes as well as any type of healthcare professional (doctors, nurses and nursing home staff) to examine their behavioural outcomes. In this review, ICT is defined as a team composed of medical and nursing staff certified infection prevention and control or its equivalent [2]. The primary and secondary outcomes of this review are as follows:

### 2.3. Primary Outcome

#### 2.3.1. Patient-Based Outcomes

Incidence rate of HCAIs (for the incidence rate of HCAIs, we did not restrict the types of HCAIs or the timing of outcome assessment. The incidence rate refers to the number of infection episodes per 1000 days or the number of infected patients per total number of patients during the study period).Death due to HCAIs (the rate of death due to HCAIs refers to the number of patients who died with HCAIs per total number of patients with HCAIs).Length of hospital stays presented in days.

#### 2.3.2. Staff-Based/Behavioural Outcomes

Compliance with infection control practices as measured by study authors.

### 2.4. Secondary Outcome 

Cost related to HCAIs.

### 2.5. Search Strategies

We searched four electronic databases, namely PubMed, EMBASE, CINAHL and Cochrane Central Register of Controlled Trials (CENTRAL), from inception to May 2020. The search strategy is presented in the protocol [18] and Appendix A). We imported the output of four databases to the EndNote reference management software [20]. Duplicate studies were removed, and the remaining ones were imported to the online reference management software, Covidence (Covidence systematic review [software], Veritas Health Innovation, Melbourne, Australia; available at www.covidence.org, accessed on 28 September 2020) for the study selection.

### 2.6. Study Selection

Five pairs of authors (MMT, MOR, RH, SM, SO, JM, YY, CM, MN and TB) independently performed screening of titles and abstracts for potential eligibility. Disagreements were resolved through discussions. Then, pairs of authors (MMT, MOR, RH, SO, JM, YY and CM) independently evaluated the full texts and selected the studies by applying the eligibility criteria (Appendix A). Disagreements were settled by discussions or involvement of the third author (TB). In the published protocol of this review, we described that we would exclude the study if the study outcomes were hand hygiene compliance or antimicrobial prescription. However, we decided to update the protocol and included the studies reporting hand hygiene among others, considering that if we strictly follow the original exclusion criteria, we will have to exclude studies that report our primary or secondary outcomes.

### 2.7. Data Extraction

The first author (MMT) prepared a standardised data extraction form and piloted the form using at least one related study, and two authors (MOR and RH) reviewed it. For each study, the following data were extracted: study author (first author only), year of publication, study design, setting, country, characteristics and number of participants, details of intervention and control, types of outcome measures and study results. The data was independently extracted by two of the authors (MMT, MOR or RH), and discrepancies between the authors were resolved through discussions.

### 2.8. Risk-of-Bias Assessment

Two of the three authors (MMT, MOR or RH) independently evaluated the risk of bias of each included study using the Cochrane risk-of-bias tool [21]. This tool addresses random sequence generation, allocation concealment, blinding of participants and personnel, blinding of outcome assessment, incomplete outcome data, selective reporting and other bias. Each domain was rated as low, high or unclear risk of bias. Any discrepancies were resolved through discussions.

### 2.9. Data Analysis

We conducted meta-analyses when two or more studies investigated a comparable intervention and outcome. We entered the raw data in RevMan 5.4 to pool the results and employed random-effect meta-analysis as most ICTs can be considered complex interventions [22]. We used the risk ratio (RR) for dichotomous data and the mean differences for continuous data with corresponding 95% confidence intervals (CI). We employed the I2 statistic to measure the heterogeneity among the included studies and interpreted the results in accordance with the definitions in the Cochrane Handbook for Systematic Reviews of Interventions [23]. Furthermore, we reported the results narratively when there were an insufficient number of studies for data to be pooled.

According to our protocol, we conducted subgroup analysis by categories of intervention by ICT (performing surveillance of HCAIs, educating healthcare professional and monitoring infection control practices). However, we could not conduct subgroup analysis by type of healthcare facilities (hospital, nursing homes, or others) due to an insufficient number of studies. We planned to conduct sensitivity analysis on primary outcomes by excluding trials with a high risk of bias. However, all the studies had a high risk of bias, so we did not conduct any sensitivity analysis.

We used the Grading of Recommendations Assessment, Development and Evaluation (GRADE) approach to evaluate the certainty of evidence for three outcomes: incidence rate of HCAIs, death due to HCAIs and compliance with infection control practices [24]. The GRADE approach has five domains (risk of bias, imprecision, inconsistency, indirectness and publication bias) to rate the certainty of evidence as very low, low, moderate and high. We were not able to evaluate publication bias through funnel plot as there were an insufficient number of studies for this assessment. We used the GRADEpro web-based platform to make a ‘Summary of Findings’ table, considering the certainty of evidence (GRADEpro GDT: GRADEpro Guideline Development Tool [Software]. McMaster University and Evidence Prime, 2022, available at www.gradepro.org, accessed on 31 May 2022).

## 3. Results

### 3.1. Results of the Search

In total, 12,666 studies were identified from the four databases. After removing duplicates, 11,719 titles and abstracts were screened. The screening excluded a large number of studies (*n* = 11,676). A total of 43 studies were assessed using the full text. Finally, 35 studies were excluded due to inappropriate study design, intervention or outcome and lack of full text (conference abstract and protocol). Nine studies were included in this review (Figure 1) [25,26,27,28,29,30,31,32,33].

### 3.2. Characteristics of the Included Studies

The characteristics of the included studies are presented in Table 1. The included studies were published between 1990 and 2020 and were cluster-randomised controlled trials. The number of clusters was small, ranging from 6 to 45. The clusters were hospital wards, outpatient long-term haemodialysis units, or nursing homes. Five studies were conducted in inpatient hospitals, one in outpatient long-term haemodialysis units and three in nursing homes. Three studies were from the USA; two from Hong Kong and one each from England, Ireland, Italy and Thailand. The study duration varied from 5 weeks to 20 months.

#### 3.2.1. Participants

The participants in the included studies were patients, residents and healthcare personnel (nurses and nursing staff). The number of patients or residents was reported in four studies: 2085 patients in hospitals and outpatient long-term haemodialysis units and 1743 residents in nursing homes. The number of healthcare personnel targeted by the intervention was reported in five studies: 1508 nurses and 333 nursing home staff. In the inpatient hospital setting, the participants were nurses in three studies, patients in one study and both of them in one study. In the outpatient haemodialysis unit setting, the participants were patients. In the nursing home setting, the participants were residents in one study and both residents and nursing home staff in two studies.

#### 3.2.2. Description of Interventions

Four studies investigated the effect of ICT without ICLN system [28,29,31,32], whereas the other five studies examined the effect of ICT with ICLN system ([25,26,27,30,33] (Appendix B Table A1). The members of ICT in the included studies were infection control doctors, ICNs, nephrologists, dialysis staff, surgical intensive care unit (ICU) co-directors, ICU physicians, nurses and infection control practitioners. The ICLN system was implemented, with selection of nurses or staff as link nurses, opinion leaders, or champions. The ICTs in the included studies performed the following infection control measures: developing and disseminating guidelines and policies [26,33], coordinating continuous education and training [25,26,27,29,30,32,33], performing surveillance of HCAIs [28,31], monitoring and auditing the practices [26,27,29,30,32,33] and standards of care and building effective links with other staff and departments [25,26,27,30,33].

#### 3.2.3. Control

Of the included studies, five had a usual care control, two had lecture control and two had no intervention as a control (Appendix B Table A1).

#### 3.2.4. Outcomes

Two studies reported patient-based outcomes; four studies, staff-based outcomes; and three studies, both outcomes (Appendix B Table A1). For patient-based outcomes, the incidence rate of HCAIs was evaluated in five studies [25,28,29,30,31], death due to HCAIs in two studies [28,31] and the length of hospital stay in one study [28]. For staff-based outcomes, nurses’ compliance with infection control practices [26,29,33], changes in infection control scores [25,27,33] and proportion of compliance with infection control guidelines at the facility level [30,32] were measured and reported. Only one study reported the cost related to HCAIs [28].

#### 3.2.5. Funding Sources

Six studies reported being funded, and out of those six, three did not declare the funding sources.

### 3.3. Risk of Bias

The overall risk-of-bias assessment is presented in Figure 2 and Figure 3. All studies were judged as having a high risk of bias due to one of the four reasons: (1) the participants and personnel were not blinded, (2) the outcome assessment was not blinded, (3) the outcome data were incomplete and (4) there was a significant imbalance between the intervention and control clusters at baseline. Blinding is inherently difficult with ICT interventions as the intervention can be easily identified by ICT members or the participants.

### 3.4. Effectiveness of Interventions

The summary of findings is presented in Table 2.

#### 3.4.1. Primary Outcomes

##### Patient-Based Outcomes

Incidence Rate of HCAIs

Five studies evaluated the incidence rate of HCAIs [25,28,29,30,31]. Three studies that reported the number of patients/residents with HCAIs were included in the meta-analysis (Figure 4). Overall, the ICT with or without ICLNs had no significant effect on the incidence rate reduction of HCAIs (RR = 0.65, 95% CI: 0.45–1.07, very low certainty of evidence). We conducted a subgroup analysis by dividing the studies according to the categories of intervention by ICT: (1) surveillance of HCAIs [28,31] and (2) continuous education and monitoring of infection control practices [25]. The subgroup analysis did not exhibit any significance in the test for subgroup differences (Group 1.1.1: RR = 0.52, 95% CI: 0.30–0.88, Group 1.1.2: RR = 0.98, 95% CI: 0.67–1.41, *p* = 0.06) (Figure 4).

Two studies reported the mean HCAI rate [29,30]. We could not conduct a meta-analysis due to the unavailability of the number of participants in the study conducted in hospitals in USA [29]. The authors in that study reported significant reduction in the mean HCAI rate in the intervention groups compared to the control groups (adjusted incidence rate ratio = 0.19, 95% CI: 0.06–0.57). Another study conducted in nursing homes in USA reported no significant reduction in the mean HCAI rate in the intervention groups compared to the control groups (relative difference = −6.7, 95% CI: −36.2–36.4) [30].

2.Death Due to HCAIs

Two studies reported death due to HCAIs [28,31]. The meta-analysis (Figure 5) revealed no significant effect of the ICT intervention on death due to HCAIs (RR = 0.32, 95% CI: 0.04–2.69, very low certainty of evidence).

3.Length of Hospital Stay

One study reported this outcome [28]. The result of this study indicated no significant difference in the length of hospital stay between the intervention and control groups (42 days vs. 45 days, *p* = 0.52).

##### Staff-Based/Behavioural Outcomes

The included studies reported compliance in three different outcome measurements.

Proportion of Compliance with Infection Control Practices

Three studies evaluated the proportion of compliance with infection control practices among nurses or staff. One study [29] could not be included in the meta-analysis as it insufficiently reported its outcome by providing only the figure without any numerical data. In that study, the infection control practices were increased in both the intervention and control groups over time [29]. The meta-analysis of two studies (Figure 6) revealed a significant effect (RR = 1.17, 95% CI: 1.00–1.38, moderate certainty of evidence) [26,33].

2.Changes in the Infection Control Compliance Score

Three studies evaluated the changes in the infection control compliance scores. One study conducted in the nursing homes reported that the mean infection control audit score was significantly higher in the intervention group than in the control group at 12 months (82% vs. 64%, *p* < 0.001) [25]. One study conducted in the hospital reported a significant increase in the self-reported compliance with standard precaution scores (15.43 vs. 14.32, *p* = 0.024) [27]. One study conducted in the hospital reported the change in the mean infection control practice scores, suggesting that the ICT with ICLNs was superior to the control (5.63 in the intervention group with lectures and demonstration, 4.96 in the intervention group with demonstration vs. 3.29 in the control, *p* < 0.05 in both comparisons) [33].

3.Proportion of Compliance with Infection Control Guidelines at the Facility Level

Two studies conducted in the nursing homes reported the compliance with infection control guidelines at the facility level. One randomised pair-matched pilot study reported the compliance at the facility level in two outcomes: weekly surface swab bacterial counts and hand-washing occasions per resident/week [30]. The two outcomes were found to be improved in the intervention in nursing homes compared with the control. One cluster-RCT reported on hand hygiene facilities, environmental cleanliness and safe disposal of clinical waste [32]. No statically significant difference was observed between the intervention and control groups.

#### 3.4.2. Secondary Outcomes

##### Cost Related to HCAIs

Only one study reported cost related to HCAIs [28]. The result of this study indicated significant difference in the cost for the treatment of HCAIs between the intervention and control groups (USD 337.3 vs. USD 516.6, *p* = 0.01).

## 4. Discussion

### 4.1. Main Findings

This systematic review synthesised the effectiveness of the ICT, with or without ICLN system, in inpatient hospital, outpatient long-term haemodialysis unit and nursing home settings. Nine cluster-RCTs included in this review tested five different types of infection control measures performed by the ICT with or without ICLN system: formulating and revising guidelines, performing surveillance of HCAIs, training and educating healthcare professionals/staff, monitoring and auditing practices and standard of care and liaising with other staff and departments. The control groups received usual care, lecture only, or no intervention. All the included studies were rated as having high risk of bias. Overall, we found no significant evidence suggesting that the ICT, with or without ICLN system, compared with the control is effective in reducing the incidence of HCAIs (very low certainty of evidence), or death due to HCAIs (very low certainty of evidence). However, we found significant evidence suggesting that the ICT with ICLN system compared with the control is effective in improving nurses’ compliance with infection control practices (moderate certainty of evidence). We were unable to conduct a meta-analysis for two outcomes, length of hospital stay and cost related to HCAIs, due to the heterogeneity of outcome measures and only one study evaluating the outcome, respectively.

Here are the possible reasons for the failure of ICT in reducing the incidence rate of HCAIs in the studies included in this review. The first reason could be the lack of surveillance for infection by ICT. Among the categories of intervention by ICT, surveillance could be a key category for prevention of infections or outbreak. Hence, surveillance used to be the primary task of infection preventionists [34,35]. The scope of surveillance depends on the type of facility and resources available; thus, each ICT should perform surveillance appropriate to its facility and resources. Second, ICT in nursing homes could face more challenges due to the fact that the nursing home setting is not a closed system [25]. The unrecognised reservoirs of infection, either residents or staff, may have remained undetected during the study due to non-participation and they may limit the success of ICT intervention in reducing the infections. Decolonisation of colonised patients and healthcare workers may be an additional effective measure to reduce HCAIs.

### 4.2. Agreements and Disagreements with Other Reviews

To the best of our knowledge, this is the first systematic review of RCTs that investigated the effectiveness of the interventions performed by the ICT, with or without ICLN system, in preventing HCAIs. Previous Cochrane reviews on professional adherence to infection control guidelines, adherence to standard precautions and infection control strategies for methicillin-resistant Staphylococcus aureus found that education alone or with other supportive interventions (employing specialised infection control personnel) may improve professional adherence to infection control guidelines [36,37,38]. We included six additional RCTs not included in the above Cochrane reviews [27,28,29,30,31,33] and excluded some studies included in the Cochrane reviews as they were not RCTs.

Another systematic review including one of our nine RCTs reported that multidisciplinary intervention, including education by the ICT, is effective in improving compliance with infection control practices among healthcare workers [39]. One systematic review by Aboelela et al. reported that educational programmes and establishment of a multidisciplinary team may be effective strategies in reducing HCAIs; however, the effects of interventions on healthcare workers’ compliance with infection control practices were mixed [40]. These findings were based on the non-randomised intervention (pre–post comparison) studies that achieved quality scores of 80% or higher.

### 4.3. Implications for Policy

ICT in inpatient setting is already recommended in many countries [41] although it may not be the case for an ICLN system. Our review showed that ICT with an ICLN system, improved nurses’ compliance with infection control practices although it does not offer any implication on the effect of ICT without an ICLN system on nurses’ compliance with infection control practices or its comparative effectiveness.

Certain barriers to the improvement of compliance with infection control practices may stand in the way of the successful implementation of ICT, such as workload, lack of manpower and insufficient time and budget as suggested by Alhumaid et.al. [42]. This highlights the need for the support of the government and leadership of the directors of healthcare facilities or legislation.

### 4.4. Implications for Research

This systematic review proves the need for higher-quality studies in the research on ICT. WHO strongly recommended that infection control program with a dedicated and trained team should be in place in acute health care facilities for the prevention of HCAIs. However, this recommendation was based on limited published evidence, only two studies [43]. This shows that the effectiveness of ICT in reducing HCAIs has not been well evaluated; thus, it is advisable to conduct more studies to provide robust insights, especially in the current COVID-19 era. In the future update of this review, it may be possible to evaluate the effects of ICT on the outbreak response during the COVID-19 pandemic. A head-to-head RCT of ICT with an ICLN system and ICT without an ICLN system should ideally be conducted to answer the question on whether an ICLN system is an indispensable component of an effective ICT.

Future studies should include both patient-based clinical outcomes (such as incidence rate of HCAIs, death due to HCAIs and length of hospital stay) and staff-based outcomes to obtain a full picture of the effectiveness of ICT. For the staff-based outcome, a better reporting of compliance of infection control practices should be applied. Only two or three studies were included in the meta-analysis of each outcome due to variation in the outcome measures or inadequate outcome data. Future studies should employ similar outcome measures to allow comparison across studies and pooling of results to provide strong evidence.

Previous studies reported that nurses’ compliance with infection control practices can help reduce the rate of HCAIs [44,45,46]. A clear link between ICT and improvement of compliance has been proven by our analysis; however, there is no clear evidence to link the improvement of compliance ICT to HCAI reduction. Further analysis is warranted to evaluate a causal relationship between improvement of compliance and subsequent reduction in HCAIs.

### 4.5. Strengths and Limitations

The strengths of our review are the comprehensive literature searches, rigorous methodology (assessment of eligibility, data extraction and assessment of risk of bias independently) and use of the GRADE approach in rating the certainty of evidence for each outcome. However, this review has some limitations. First, the detailed information of each ICT could not be obtained from the included studies, such as how the ICT was developed and the members of ICT were selected in each included study. Second, the interventions and outcome measurements were heterogeneous. There were differences in the format, structure and length of interventions delivered by the ICT and ICLNs. A high variability in outcome measurement rendered pooling of the result data inappropriate. Third, the certainty of evidence was very low as a consequence of high risk of bias, heterogeneity across the studies and imprecision of the meta-analyses results. Fourth, we restricted this review to RCTs, and found only a small number of heterogeneous studies with low-quality evidence. Fifth limitation is the old publication year of some of the studies that may not reflect the current infection control resources or the challenges.

## 5. Conclusions

There is a dearth of evidence for the effect of ICT, with or without ICLN system, in inpatient hospitals and outpatient and long-term care facilities. We did not observe any statistically significant evidence of ICT in reducing the incidence rate of HCAIs, death due to HCAIs and length of hospital stay; however, ICT with ICLN system likely improves nurses’ compliance with infection control practices. Due to the high level of bias, inconsistency and imprecision, these findings should be considered with caution. High-quality studies using similar outcome measures are needed to demonstrate the effectiveness and cost-effectiveness of ICT in the healthcare settings.

## Figures and Tables

**Figure 1 ijerph-19-17075-f001:**
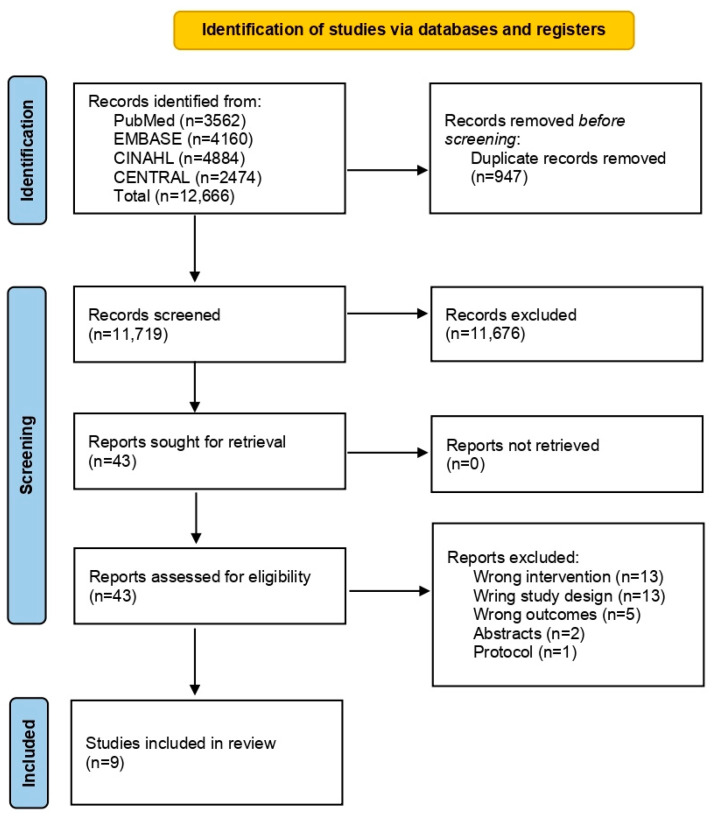
PRISMA flow diagram of literature search, screen and selection criteria.

**Figure 2 ijerph-19-17075-f002:**
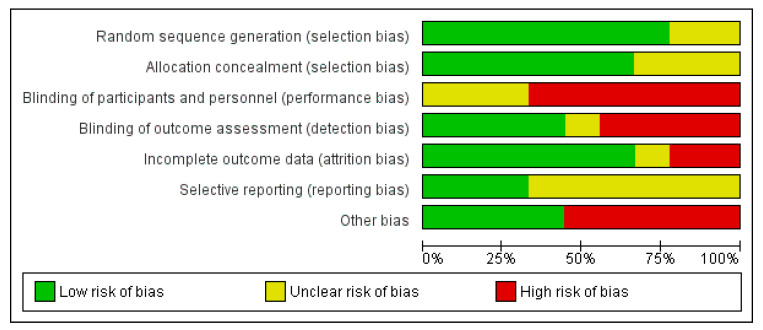
Risk of bias graph: review authors’ judgements about each risk of bias item presented as percentages across all included studies.

**Figure 3 ijerph-19-17075-f003:**
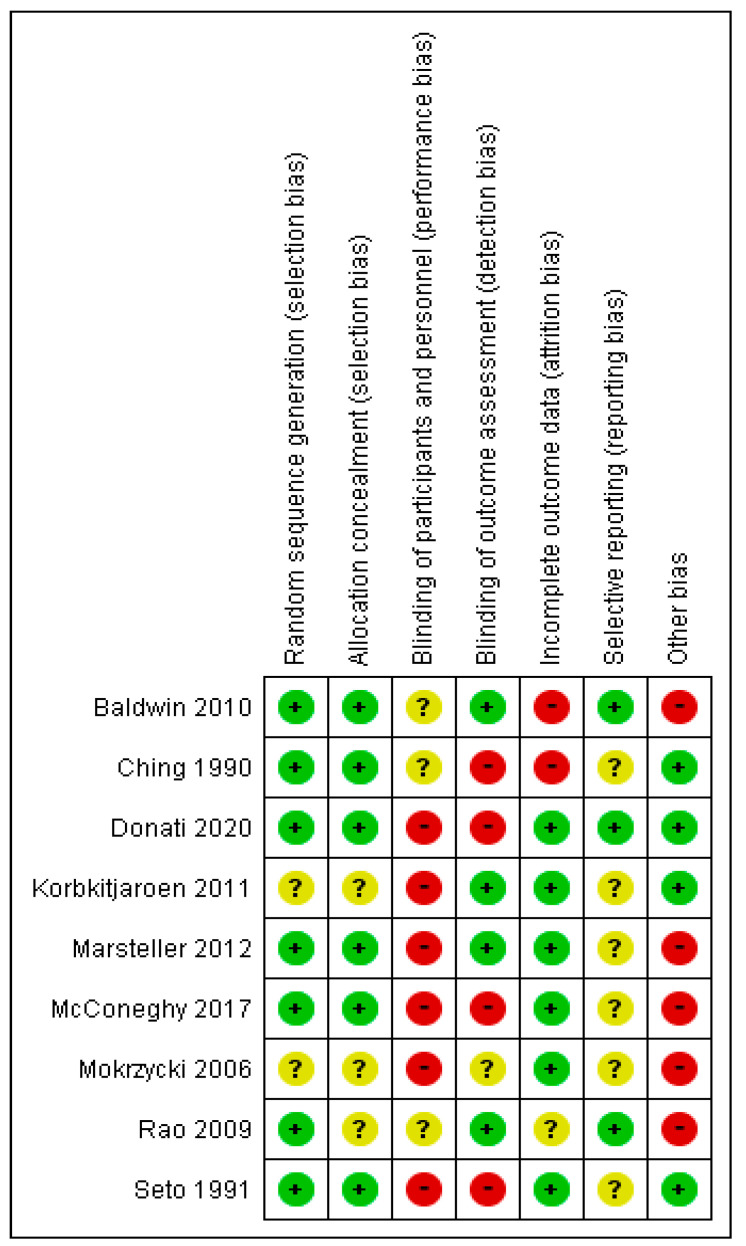
Risk of bias summary: review authors’ judgements about each risk of bias item for each included study [[25],[26],[27],[28],[29],[30],[31][32],[33],].

**Figure 4 ijerph-19-17075-f004:**
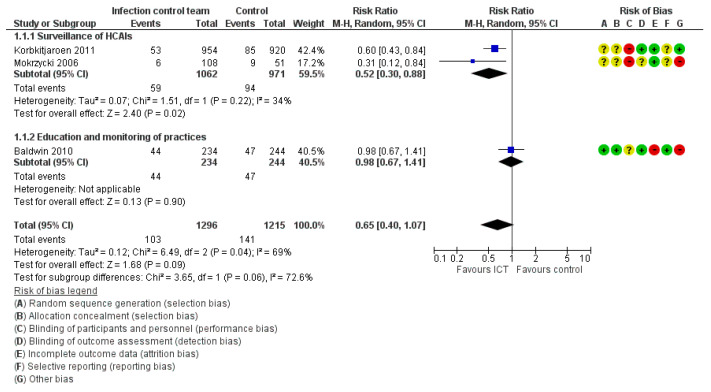
Meta-analysis for the incidence rate of HCAIs [25,28,31].

**Figure 5 ijerph-19-17075-f005:**
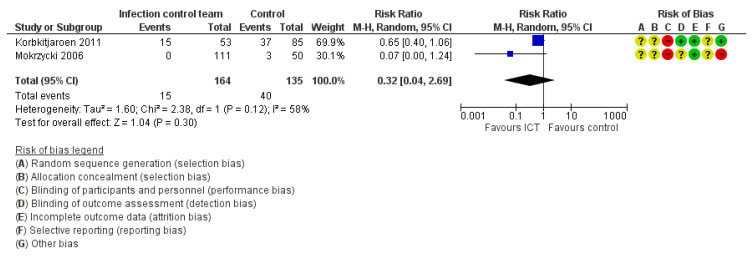
Meta-analysis for the death due to HCAIs [28,31].

**Figure 6 ijerph-19-17075-f006:**
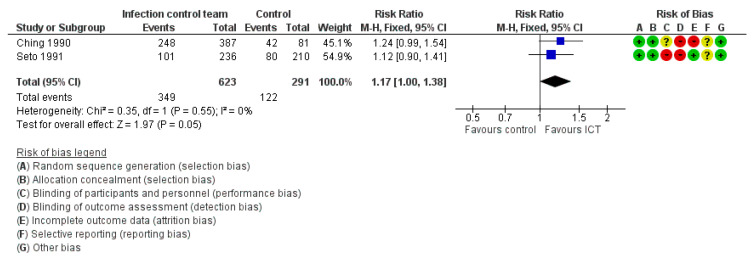
Meta-analysis for the proportion of compliance with infection control practices [26,33].

**Table 1 ijerph-19-17075-t001:** Characteristics of included studies (*n* = 9).

Characteristics		No	%
Publication year	1990–2000	2	22.22
	2001–2010	3	33.33
	2011–2020	4	44.44
Location	USA	3	33.33
	Europe	3	33.33
	Asia	3	33.33
Setting	Inpatient hospitals	5	55.56
	Outpatient haemodialysis units	1	11.11
	Nursing homes	3	33.33
Type of intervention	ICT	4	44.44
	ICT + ICLN system	5	55.56
Outcome assessed	Patient-based		
	HCAIs	5	55.56
	Deaths	2	22.22
	Length of hospital stay	2	22.22
	Staff-based		
	Compliance	7	77.78
	Cost	1	11.11

HCAIs, healthcare-associated infections; ICLN, infection control link nurse; ICT, infection control team; USA, The Unite States of America.

**Table 2 ijerph-19-17075-t002:** Summary of findings and GRADE evidence profile.

Outcomes	Anticipated Absolute Effects (95% CI)		Relative Effect (95% CI)	No of Participants (Studies)	Certainty of the Evidence (GRADE)
	Risk with Usual Care	Risk with Infection Control Team			
Incidence rate of HCAIs (follow-up: range 4 months to 20 months)	116 per 1000	75 per 1000 (46 to 124)	RR 0.65 (0.40 to 1.07)	2511 (3 RCTs)	⨁◯◯◯ Very low ^a,b,c^
Death due to HCAIs (follow-up: range 4 months to 20 months)	296 per 1000	95 per 1000 (12 to 797)	RR 0.32 (0.04 to 2.69)	299 (2 RCTs)	⨁◯◯◯ Very low ^a,b,c^
Compliance with infection control practices (follow-up: mean 5 weeks)	419 per 1000	491 per 1000 (419 to 579)	RR 1.17 (1.00 to 1.38)	914 (2 RCTs)	⨁⨁⨁◯ Moderate ^a^

Explanations: ^a^ Downgraded one level due to performance bias, attrition bias and other bias; ^b^ Downgraded one level for inconsistency due to heterogeneity across the studies (I2 > 50%); ^c^ downgraded one level for imprecision due to wide 95% CI.

## Data Availability

Not applicable.

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
