# Peer review of "Effectiveness of Infection Control Teams in Reducing Healthcare-Associated Infections: A Systematic Review and Meta-Analysis"

_ijerph, 2022, doi:10.3390/ijerph192417075_

Round 1

Reviewer 1 Report

This systematic review aimed to evaluate the effectiveness of ICT, with 15 or without an infection control link nurse  system, in reducing healthcare associated infections. And quantitative analysis is carried out with the method of meta analysis. In general, the completion of this paper is quite good, except for some minor problems that need to be corrected.

1. Please supplement the specific search formula.

2. The search date of the literature is up to May 2020. As this is a systematic review and meta-analysis, it is recommended that the author update the literature.

3. Considering the impact of the COVID-19 on hospital infection, the current topic cannot cover the actual work status of the infection control team, so I suggest that the topic be modified to exclude the impact of the COVID-19.

4. Table 1, what is the meaning of "No" and "%"? I had a hard time figuring out what they meant. Are these percentages necessary?

Author Response

1. Please supplement the specific search formula.

Thank you for your comments. We have added the search strategy as a supplementary file (Supplementary Material Table S3).  

2. The search date of the literature is up to May 2020. As this is a systematic review and meta-analysis, it is recommended that the author update the literature.

We will be updating the literature search and subsequently updating the review in the future.  

3. Considering the impact of the COVID-19 on hospital infection, the current topic cannot cover the actual work status of the infection control team, so I suggest that the topic be modified to exclude the impact of the COVID-19.

The protocol for this review was developed prior to the COVID-19 period. We will modify the exclusion criteria in the future update.

4. Table 1, what is the meaning of "No" and "%"? I had a hard time figuring out what they meant. Are these percentages necessary?

To summarize the characteristics of the included studies, we used "no" and "%" in Table 1. Percentages were obtained by dividing the number of studies on a particular characteristic by the total number of included studies.

Reviewer 2 Report

The authors address a topic of considerable importance for improving the quality of care. The role of nurses now seems indispensable, and the authors have carried out a good systematic review and a rather original meta-analysis. The statistical methodology is correct and satisfies the objectives of the study. The results and tables are clear.

Author Response

Thank you for your comments.

Reviewer 3 Report

The introduction is scarce, it would be necessary to give more details on the TCI and ICLN, since when they have been implemented and in which countries.

It is not clear the inclusion criteria, could some exclusion criteria also be included? It is not explained why so many studies were excluded, it is only explained in the 43 studies that were excluded, but it would be necessary to better identify the inclusion and exclusion criteria to justify only 9 studies that make up the meta-analysis.

The results found are very old, more than half of those found are studies more than 10 years old, so the weight of this study is very low.

In the section on participants it is not clear the interventions of the nurses and the patients, it should be better explained especially in the residences, the interventions were performed on the nurses and the patients benefited were ???

The methodology used I believe is correct.

Conclusions are scarce, although with only 9 studies, not much more can be concluded. 

Author Response

1. The introduction is scarce, it would be necessary to give more details on the TCI and ICLN, since when they have been implemented and in which countries.

Thank you for your comments. We have added the following sentence in the last paragraph of the “Introduction”:

“ICT was established in the United Kingdom in 1950s and ICLN system was first introduced in 1990s.”

2. It is not clear the inclusion criteria, could some exclusion criteria also be included? It is not explained why so many studies were excluded, it is only explained in the 43 studies that were excluded, but it would be necessary to better identify the inclusion and exclusion criteria to justify only 9 studies that make up the meta-analysis.

The eligibility criteria were added as a supplementary file (Supplementary Material Table S2).

3. The results found are very old, more than half of those found are studies more than 10 years old, so the weight of this study is very low.

Thank you for your comment. Although some of the studies are more than 10 years old, some included studies are newer and no systematic review of this scope has ever been published, which makes the finding new and original. Therefore, we believe that our study adds important knowledge to this field.

4. In the section on participants it is not clear the interventions of the nurses and the patients, it should be better explained especially in the residences, the interventions were performed on the nurses and the patients benefited were ???

The eligible participants in this review were the patients of the health care facility or the residents in the community, and healthcare professionals at the health care facilities where the intervention took place (for patient-based outcomes) and (for staff-based outcomes). The interventions were performed on the healthcare professionals, and the benefited persons in the included studies were the patients in inpatient and outpatient facilities, residents in long-term care facilities and the healthcare professionals (mainly nurses and nursing staff). We have addressed according to your comment in “2.2 Eligibility criteria” as follow:

“We included studies on patients of all ages in inpatient hospitals and outpatient facilities, and residents in long-term care facilities to examine the patient-based outcomes as well as any type of healthcare professional (doctors, nurses and nursing home staff) to examine their behavioural outcomes.”

We also revised in “3.2.1 Participants” as follows:

“The participants in the included studies were patients, residents and healthcare personnel (nurses and nursing staff). The number of patients or residents was reported in four studies: 2,085 patients in hospitals and outpatient long-term haemodialysis units and 1,743 residents in nursing homes. The number of healthcare personnel targeted by the intervention was reported in five studies: 1,508 nurses and 333 nursing home staff. In the inpatient hospital setting, the participants were nurses in three studies, patients in one study and both of them in one study. In the outpatient haemodialysis unit setting, the participants were patients. In the nursing home setting, the participants were residents in one study and both residents and nursing home staff in two studies.”

5. The methodology used I believe is correct.
Conclusions are scarce, although with only 9 studies, not much more can be concluded.

Thank you for your comment. We conducted rigorous search and careful review with clear criteria. Therefore we believe that our review provides the best knowledge among all the published studies so far on this important topic. We are planning to conduct a review for the healthcare settings providing COVID-19 care.

Reviewer 4 Report

Dear authors,

the manuscript is well written and contains all the parts that are important for a quality article. In your review, you have applied a rigorous methodology for which you have clearly stated additional limitations that will certainly be useful for some researchers in the future. I have no further suggestions to improve your manuscript.

Author Response

Thank you for your comments.